# Post-click labeling enables highly accurate single cell analyses of glucose uptake ex vivo and in vivo
Masaki Tsuchiya[1,2,3], Nobuhiko Tachibana[1,2] & Itaru Hamachi [1,4] ✉

Cellular glucose uptake is a key feature reflecting metabolic demand of cells in physiopathological conditions. Fluorophore-conjugated sugar derivatives are widely used for monitoring glucose transporter (GLUT) activity at the single-cell level, but have limitations in in vivo applications. Here, we develop a click chemistry-based post-labeling method for flow cytometric measurement of glucose uptake with low background adsorption. This strategy relies on GLUT-mediated uptake of azide-tagged sugars, and subsequent intracellular labeling with a cell-permeable fluorescent reagent via a copper-free click reaction. Screening a library of azide-substituted monosaccharides, we discover 6-azido-6-deoxy-D-galactose (6AzGal) as a suitable substrate of GLUTs. 6AzGal displays glucose-like physicochemical properties and reproduces in vivo dynamics similar to $^{18}$F-FDG. Combining this method with multi-parametric immunophenotyping, we demonstrate the ability to precisely resolve metabolically-activated cells with various GLUT activities in ex vivo and in vivo models. Overall, this method provides opportunities to dissect the heterogenous metabolic landscape in complex tissue environments.

Cellular glucose uptake is crucial for physiological and pathological processes, and mediated by the glucose transporter (GLUT) family[1,2]. GLUT expression on the cell surface facilitates glucose influx to satisfy energy demand in metabolically active cells, such as cancer and immune cells. Because cellular metabolism is intrinsically dynamic and heterogenous in tissues, assays to determine glucose transport activities in individual cells are important and urgently needed[3–5]. Various sugar analogs have been developed to measure glucose uptake[6] (Fig. S1). 2-Deoxy-D-glucose (2DG) and its radiolabeled forms ($^{3}$H-2DG and $^{18}$F-FDG) passing through GLUTs provide a reliable readout in bulk measurements but lack cellular resolution[6–9]. To monitor glucose transport at the single cell level, flow cytometry- and microscopy-based approaches using fluorescent sugar analogs are commonly employed[6]. However, conjugation of a fluorophore to glucose has undesirable effects on its properties and interactions with GLUTs[4,6,9–14]. For example, 2NBDG and Cy5.5-2DG (Fig. S1) are fluorescent 2DG derivatives with larger molecular sizes (342 and 1089 Da, respectively) than glucose (180 Da) and do not properly reproduce natural GLUT-

dependent glucose influx, causing non-specific background staining in cells and tissues[4,6,9–14]. This shortcoming substantially hampers accurate single cell analysis of glucose uptake ex vivo and in vivo.

Here, we developed a click chemistry-based post-labeling method for flow cytometric high-throughput measurement of glucose uptake with minimal perturbation of GLUT activity and low non-specific cellular adsorption. This strategy (Fig. 1) relies on GLUT-mediated uptake of a clickable azide-tagged sugar, and subsequent intracellular labeling with a cell-permeable fluorescent reagent (BDP-DBCO, Fig. S2a) via a copper-free click reaction[15,16]. By screening a library of azide-substituted monosaccharide isomers, we discovered and validated 6-azido-6-deoxy-D-galactose (6AzGal) as a suitable substrate for GLUTs. 6AzGal displays glucose-like physicochemical properties and reproduces in vivo dynamics similar to $^{18}$F-FDG. Combining this method with multi-parametric immunophenotyping, we demonstrated the ability to precisely resolve metabolically-activated cells with various glucose transport activities in ex vivo and in vivo models.

[1]Department of Synthetic Chemistry and Biological Chemistry, Graduate School of Engineering, Kyoto University, Katsura, Nishikyo-ku, Kyoto 615-8510, Japan. [2]PRESTO (Precursory Research for Embryonic Science and Technology, JST), Sanbancho, Chiyoda-ku, Tokyo 102-0075, Japan. [3]School of Pharmaceutical Sciences, University of Shizuoka, 52-1 Yada, Suruga-ku, Shizuoka 422-8526, Japan. [4]ERATO (Exploratory Research for Advanced Technology, JST), Sanbancho, Chiyoda-ku, Tokyo 102-0075, Japan. ✉e-mail: ihamachi@sbchem.kyoto-u.ac.jp

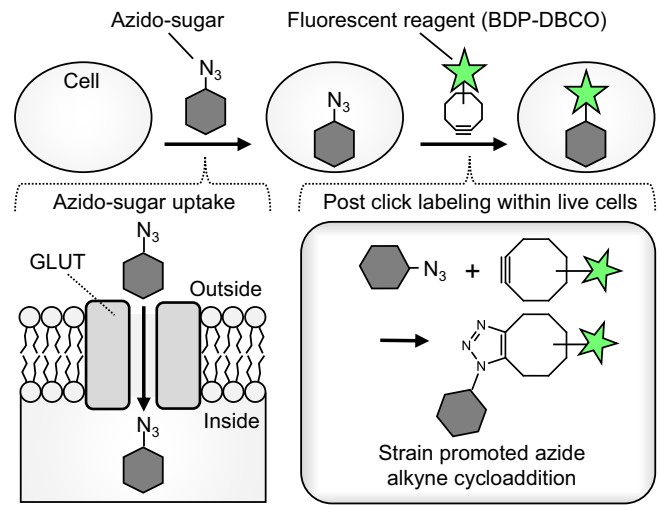

**Fig. 1 | Schematic for measurement of azido-sugar uptake by post-click labeling.** Azido-sugars are taken up through GLUTs into cells and subsequently labeled with BDP-DBCO via a click reaction. After washout of unreacted BDP-DBCO, cells containing BDP-DBCO-labeled azido-sugars are detected by fluorescence microscopy or flow cytometry.

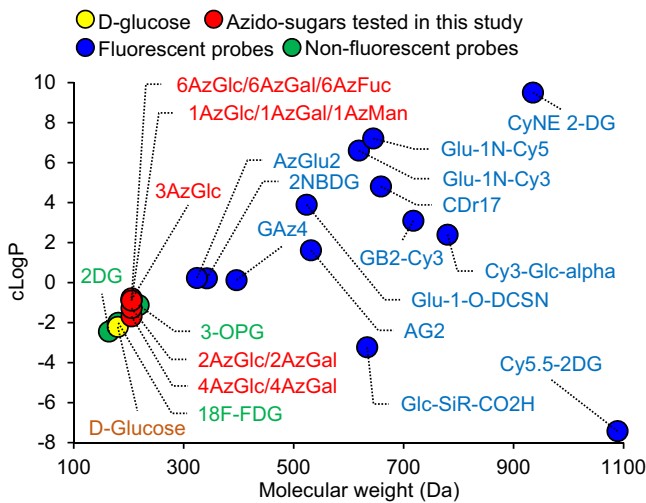

**Fig. 2 | Physicochemical properties of D-glucose and its analogs.** Molecular weight and cLogP were calculated by ChemDraw (chemical structures shown in Figs. S1 and S2b).

## Results

### Development of post-click labeling method for glucose uptake assay

Considering that GLUTs are known to import various sugars, including glucose, galactose, and their derivatives[6,7,13,14,17], these minimally modified analogs may be recognized as substrates of GLUTs. A series of monosaccharides with a single azide substitution (Fig. S2b) maintained a low molecular weight (205 Da) comparable with glucose (180 Da) (Fig. 2). Hydrophilicity of these azido-sugars (evaluated by their cLogP value: −2 to −1) was in a similar range to that of glucose (cLogP: −2) (Fig. 2). Additionally, their physicochemical properties were almost identical to well-validated non-fluorescent probes 2DG[7], [18]F-FDG[9,14], and 3-OPG[13], outperforming the existing fluorescent analogs (Fig. 2 and S1). To measure azido-sugar uptake (Fig. 1), cells were incubated with azido-sugars, followed by BDP-DBCO treatment and subsequent washing to remove unreacted BDP-DBCO. Of 11 azido-sugars tested by flow cytometry, 6-azido-6-deoxy-D-galactose (6AzGal) produced the highest fluorescence intensity in cellular BDP-DBCO labeling (Figs. 3a, b and S2b, c). The blue-emitting BDP-DBCO

variant (8AB-DBCO)[16] also showed fluorescent labeling of 6AzGal-treated cells (Fig. S2d, e). Intense fluorescent signals of BDP-DBCO in 6AzGal-treated cells were predominantly observed in the cytoplasm by confocal microscopy with minimal fluorescence owing to cell surface glycosylation (Fig. 3c and S2f). Conversely, non-azide-treated cells exhibited much weaker fluorescent signals, representing efficient removal of unreacted BDP-DBCO, as we reported previously[16]. Fluorescent labeling after 6AzGal uptake was confirmed in three cell lines (K562, HL60S, and HCC1806) (Figs. 3b, c and S2g). No significant toxicity was detected by 6AzGal treatment or BDP-DBCO labeling (Fig. S2h). At room temperature, which is standard for the 2DG uptake assay[7], a concentration-dependent linear increase in 6AzGal uptake was observed (Fig. S2i). The uptake kinetics of 6AzGal showed a linear increase at 0–30 min and reached a plateau at 30–60 min (Fig. S2j), which was consistent with those of 3-OPG reported previously[13]. Lowering the cellular temperature blocked 6AzGal uptake (Fig. S2k), indicating transporter-dependent 6AzGal influx.

To confirm that 6AzGal passed through GLUTs, we conducted the following experiments. A competition assay showed that 6AzGal uptake was dose-dependently suppressed by D-glucose and 2DG, whereas only a minimal effect was observed using a high concentration of L-glucose, which is not recognized by GLUTs[13,14] (Fig. 3d). Cytochalasin B and WZB-117 (endofacial and exofacial GLUT inhibitors, respectively[13,14]) blocked 6AzGal uptake (Fig. 3e, f). Efflux from 6AzGal-loaded cells was also reduced by D-glucose and cytochalasin B (Fig. S2l). Using differentiated 3T3-L1 adipocytes, in which insulin stimulation promotes GLUT4 expression[14], 6AzGal uptake was increased in response to insulin (Fig. S2m). These data demonstrated that 6AzGal can act as a substrate of GLUTs.

We further examined the accuracy of this method by comparison with the enzymatic bulk 2DG uptake assay, which is the gold standard to measure glucose transport activity[7]. Output response upon 2DG uptake in the presence of a GLUT inhibitor was quantitatively evaluated. Cytochalasin B and WZB-117 treatments resulted in potent inhibitory effects on 2DG uptake (>70% signal reduction) (Fig. S2n), which was almost identical to the corresponding 6AzGal data (Fig. 3e, f). Conversely, a flow cytometric 2NBDG uptake assay showed rather weak effects of both inhibitors (only 20–40% signal reduction) (Fig. S2o), indicating non-specific cellular binding of 2NBDG as shown in previous reports[9–11,13,14]. These results demonstrated that our method achieved highly accurate single cell measurements of glucose uptake with low-background adsorption.

### Measurement of 6AzGal uptake ex vivo and in vivo

BDP-DBCO emitted bright green fluorescence with minimal overlap in orange-to-red emission (Fig. S3) and can be used for multicolor flow cytometric assays. Additionally, our click labeling of 6AzGal with BDP-DBCO never needs any other chemicals such as copper which is known to quench commonly used fluorescent proteins [e.g., mCherry and phycoerythrin (PE)]. These features made our method compatible with multiparametric phenotyping of tissue-derived cells together with lineage identification. To analyze immune cells ex vivo, leukocytes were prepared from mouse spleens, loaded with 6AzGal, and stained with surface marker antibodies and a viability dye (FVD), followed by BDP-DBCO treatment (Fig. 4a and S4a). Additional fluorophore-conjugated antibodies (e.g., AF647) and a cell-tracking dye (CPM) were optionally used for barcoding in multiplexed samples to increase throughput and decrease noise (Fig. S4b, c). The labeled splenocytes were separated into B and T cell populations by multiple gating (Fig. 4b, left and middle, and S4b). This assay showed that 6AzGal uptake was higher in B cells than in T cells (Fig. 4b, right), which was verified by a 2DG assay (Fig. S4d). Furthermore, upon T cell receptor stimulation, CD4[+] T cells expressing activation marker CD69 showed a corresponding increase in 6AzGal uptake (Figs. 4c and S4e–g), which was consistent with the previous [3]H-2DG-based observation of increased GLUT1 activity in metabolically reprogrammed T cells[8].

Next, we applied this method to in vivo analyses. We injected 6AzGal into mice intraperitoneally and isolated tissues, followed by multiple labeling similar to ex vivo experiments (Fig. 4a). In all tested tissues (spleen,

**Fig. 3 | 6AzGal acting as a substrate of GLUTs.**
**a** 6AzGal structure. **b** Quantification of 6AzGal
uptake. K562 cells were incubated with 10 mM
6AzGal in glucose-free medium for 1 h at 25 °C,
treated with 100 nM BDP-DBCO, washed with 10%
serum-containing medium, and analyzed by flow
cytometry to measure the median fluorescence
intensity (MFI). **c** Imaging of 6AzGal uptake.
6AzGal-treated HCC1806 cells were labeled with
BDP-DBCO and analyzed by confocal microscopy. **d**
Competitive inhibition of 6AzGal uptake by
D-glucose and 2DG, but not L-glucose. K562 cells
treated with 10 mM 6AzGal in the presence of the
indicated sugars were labeled with BDP-DBCO and
analyzed by flow cytometry (Left: representative
histogram. Right: bar graph based on BDP-DBCO
MFI differences). **e, f** Blocking 6AzGal uptake by
cytochalasin B and WZB-117. K562 and HCC1806
cells treated with 10 mM 6AzGal in the presence of
the indicated GLUT inhibitors were analyzed by
flow cytometry (**e**) and confocal microscopy (**f**),
respectively. Bar graphs represent means ± SEM.
*P* values were determined by the *t* test.

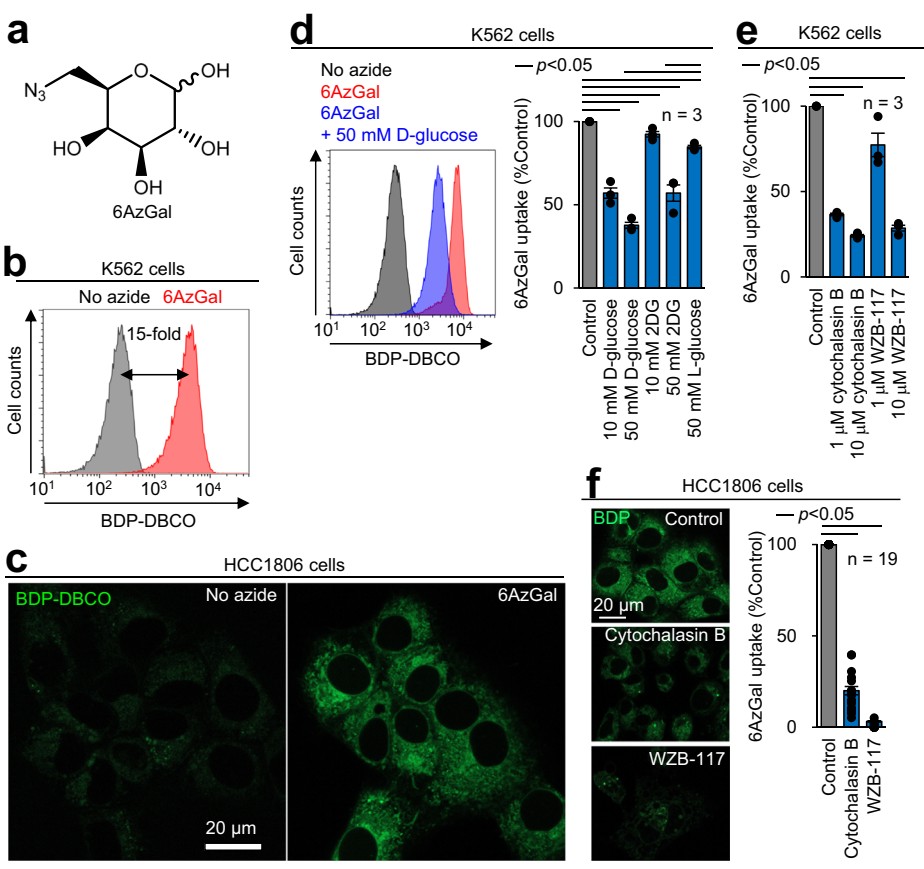

blood, thymus, and bone marrow), fluorescent labeling signals were
stronger after 6AzGal treatment compared with the no azide control
(Figs. 4d and S5a), indicating that 6AzGal underwent widespread diffusion
inside the body, followed by cellular uptake sufficient for flow cytometric
detection. The labeling levels in splenocytes strictly depended on the 6AzGal
concentration (Fig. S5b) and reached a peak at 30 min after 6AzGal injec-
tion, followed by a steady decline (Fig. S5c). Our observations of the in vivo
6AzGal distribution and kinetics agreed with previous [18]F-FDG PET ima-
ging studies in live mice[18]. Noticeably, splenic B and T cells loaded with
6AzGal in vivo showed labeling patterns (Figs. 4e and S5d) similar to the
corresponding ex vivo data of 6AzGal and 2DG (Figs. 4b and S4d). Addi-
tionally, D-glucose supplementation[14] significantly reduced the labeling
level in vivo (Fig. 4f), which was consistent with the in vitro competition
assay (Fig. 3d).

Lastly, three disease-associated mouse models were analyzed. In
response to lipopolysaccharide (LPS)-induced inflammation, which is well
known to increase glucose uptake in activated leukocytes[19], splenic B and
T cells displayed higher 6AzGal uptake than the no LPS control (Figs. 4g and
S6a, S6b). In brain immune cells isolated from mice with ischemic stroke
injury, which activates a cascade of inflammatory processes[20], we observed
increases in subpopulations with high 6AzGal uptake (Fig. S6c, d). Fur-
thermore, in K562 tumor xenografts, co-injection of 6AzGal with the GLUT
inhibitor[14] significantly reduced the uptake signal in cancer cells (Fig. 4h and
S6e), as observed in the in vitro GLUT specificity test (Fig. 3e). Overall, these
results demonstrated that this method can reliably measure glucose uptake
in vitro, ex vivo, and in vivo, and highlighted its robust utility in pathological
and pharmacological investigations.

## Discussion
We described a post-labeling method for quantitative analysis of glucose
uptake with single cell resolution. This technique provided a fluorescence
signal output which was proportional to the amount of BDP-DBCO-

labeled azido-sugars inside the individual cell. We found that 6AzGal
which possesses an azido group at the C-6 position produced the robust
signal reflecting the GLUT activity (Figs. 3 and 4), which agrees with
previous studies showing that substitution at the C-6 hydroxyl group
with a hydrophobic group enhances the recognition of the galactose
analogs by GLUTs[6,21,22]. Given that GLUT1 is reported to be the major
pathway for glucose transport in HCC1806 cells[23], our inhibition assays
(Fig. 3f) indicated that 6AzGal was transported into cells through
GLUT1, which is consistent with the fact[17] that galactose acts as a sub-
strate of GLUT1. Considering on sugar concentration in culture medium
(~25 mM) and low affinity of GLUT1 for sugar substrates ($K_m$ in the mM
range)[13], our standard incubation condition (10 mM 6AzGal for 1 hour at
room temperature) can produce an intracellular 6AzGal concentration in
the mM range. Based on our previous estimation of labeling reagents
accumulated inside cells[15], 100 nM BDP-DBCO in medium would give
the intracellular concentration of high μM to sub mM, ensuring strain-
promoted alkyne-azide cycloaddition (SPAAC) between BDP-DBCO
and 6AzGal inside cells (Fig. 1). In addition, 6AzGal has the primary
azide at the less crowed C-6 position which can accelerates the click
labeling reaction with BDP-DBCO (Fig. 3a and S2c)[21]. Since we pre-
viously identified BDP-DBCO as the most sensitive probe for organelle-
selective labeling of azide-tagged phosphatidylcholine (PC)[16], we chose
BDP-DBCO in this work and detected most of the BDP-DBCO-labeled
6AzGal in the endoplasmic reticulum and the Golgi apparatus (Fig. 3c).
This observation together with our previous PC imaging analysis[15] raised
an interesting possibility of tracing intracellular azido-sugar dynamics in
live cells, while 6AzGal underwent changes in its physicochemical
properties upon BDP-DBCO labeling (MW: from 205.2 to 755.6, cLogP:
from −0.78 to 2.5). Although the scope of this work was focused on
validating flow cytometric 6AzGal uptake assay, other DBCO reagents[16]
such as mitochondria-targeting Cy3-DBCO might have a potential to
expand organelle-selective azido-sugar labeling.

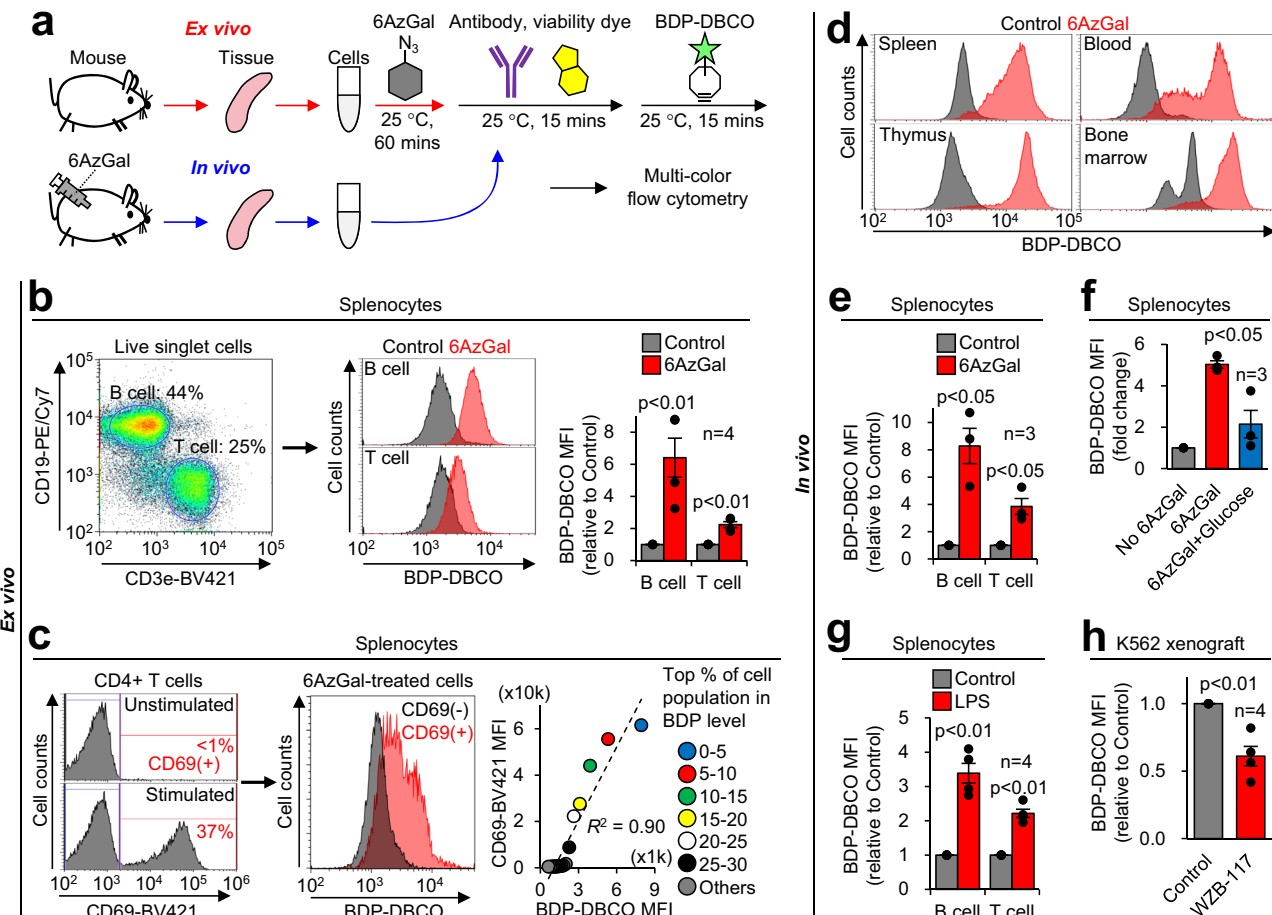

**Fig. 4 | Measurement of 6AzGal uptake ex vivo and in vivo. a** Workflow for multicolor phenotyping of cells receiving 6AzGal ex vivo and in vivo. For ex vivo experiments, cells isolated from tissues were incubated with 6AzGal, stained with antibodies and a viability dye, labeled with BDP-DBCO, and analyzed by flow cytometry. For in vivo experiments, 6AzGal was injected into mice, followed by cell isolation, multiple labeling, and flow cytometric analysis. **b** Ex vivo 6AzGal uptake in splenocytes. Cells isolated from mouse spleens were treated with 6AzGal, labeled with anti-CD marker antibodies, FVD780, and BDP-DBCO, and analyzed by flow cytometry to quantify 6AzGal labeling in B and T cells (gating strategy shown in Fig. S4b). **c** Increased 6AzGal uptake in ex vivo-activated T cells. CD3/CD28-stimulated CD4[+] T cells were analyzed by flow cytometry to quantify CD69 expression and 6AzGal labeling (gating strategy shown in Fig. S4f). Their correlation was evaluated by the order of BDP-DBCO MFI in subdivided populations (in right panel, one dot represents the MFI of $1 \times 10^3$ cells). **d** In vivo 6AzGal uptake in mouse tissues. After

6AzGal injection into mice, cells isolated from the indicated tissues were labeled with BDP-DBCO and analyzed by flow cytometry (Fig. S5a). **e** In vivo 6AzGal uptake in splenocytes. Splenocytes isolated from 6AzGal-injected mice were subjected to flow cytometric quantification of 6AzGal labeling (Fig. S5d). **f** Competitive inhibition of in vivo 6AzGal uptake by D-glucose supplementation. Splenocytes isolated from mice cotreated with 6AzGal and D-glucose were subjected to flow cytometric quantification of 6AzGal labeling. **g** Increased 6AzGal uptake in inflammation models. After 6AzGal injection into LPS-treated mice, splenocytes were isolated and subjected to flow cytometric quantification of 6AzGal labeling (Fig. S6b). **h** Blocking 6AzGal uptake in tumors by the GLUT inhibitor. Nude mice with subcutaneous K562 xenografts were coinjected with 6AzGal and WZB-117, and subjected to flow cytometric quantification of 6AzGal labeling (Fig. S6e). Bar graphs represent means ± SEM. P values were determined by the t test.

Unlike glucose-based probes (e.g., 2NBDG, $^3$H-2DG, $^{18}$F-FDG) (Fig. S1), 6AzGal has the 6-deoxygalactose backbone (Fig. 2a) which is not phosphorylated at the 6-position by hexokinase[24]. Galactokinase phosphorylates natural galactose at the 1-position, but cannot recognize 6AzGal as a suitable substrate[25–27]. Consistent with these findings, our efflux data (Fig. S2l) suggest that 6AzGal taken up by cells is not phosphorylated and eventually exported out of the cells. Given that the hexokinase-dependent phosphorylation is critical for intracellular accumulation of glucose-based probes[28,29] but not for that of 6AzGal, it is reasonable that an uptake signal of 6AzGal does not perfectly match those of glucose-based probes. Also, different biological situations (e.g., in vivo vs. ex vivo or fasting vs. non-fasting) may have impacts on the insufficient matching (please see Peer Review File for detail). The more in-depth in vivo studies may be required to explore this point using a combination of 6AzGal-based flow cytometry and $^{18}$F-FDG-based PET imaging/gamma emission recording[9,30].

This method enables a simple low-background glucose uptake assay in single cells with high accuracy comparable with the well-established bulk

2DG technique. Our method fully meets the increasing need for rapid, sensitive, and high-throughput measurements of cellular glucose transport activities ex vivo and in vivo by conventional flow cytometry[3,4], and thus substantially expands the scope of single cell biology, especially in immunometabolism and cancer fields. Our method allows multicolor, live single cell phenotyping, which makes it compatible with various applications such as lineage identification, cell sorting, RNA-seq, and CRISPR screens. With the potential to combine omics technologies[5], our method will be beneficial to dissect the heterogenous metabolic landscape in complex tissue environments.

## Methods
### Chemical reagents and antibodies
All chemical reagents from commercial suppliers were used without any further purification: 1-azido-1-deoxy-β-D-glucopyranoside (1AzGlc; Sigma, 514004), 2-azido-2-deoxy-D-glucose (2AzGlc; Sigma, 712795), 3-azido-3-deoxy-D-glucopyranose (3AzGlc; synthose, AG915), 4-azido-4-

deoxy-D-glucose (4AzGlc; synthose, AG397), 6-azido-6-deoxy-D-glucose (6AzGlc; Sigma, 712760), 1-azido-1-deoxy-β-D-galactopyranoside (1AzGal; Sigma, 513989), 2-azido-2-deoxy-D-galactose (2AzGal, Biosynth, MA03562), 4-azido-4-deoxy-D-galactose (4AzGal, synthose, AL788), 6-azido-6-deoxygalactose (6AzGal; Sigma, 712752), α-D-mannopyranosyl azide (1AzMan; synthose, MM947), 6-azido-L-fucose (6AzFuc; synthase, AF415), 2-deoxy-D-glucose (2DG; TCI, D0051), 2NBDG (Wako, 334-00631), D-glucose (Wako, 043-31163), L-glucose (Wako, 599-20963), cytochalasin B (Wako, 030-17551), WZB-117 (Sigma, SML0621), 7-diethylamino-3-(4-maleimidophenyl)-4-methylcoumarin (CPM; Wako, 045-29131), BDP FL DBCO (BDP-DBCO; BroadPharm, BP-23473), and eBioscience™ Fixable Viability Dye eFluor™ 780 (FVD780; Invitrogen, 65-0865-14). Following fluorophore-conjugated monoclonal antibodies were used for flow cytometry (1:100 dilution): CD19-PE/Cy7 (BioLegend, 115519), CD3e-BV421 (BioLegend, 100335), CD3e-BV711 (BioLegend, 100349), CD4-PE/Cy7 (BioLegend, 100527), CD4-PE/Dazzle594 (BioLegend, 100565), CD8a-SVB515 (BioRAD, MCA609SBV515), CD69-BV421 (BioLegend, 104527), CD45-AF647 (BioLegend, 103123), CD45-BV421 (BioLegend, 103133), CD11b-BV421 (BioLegend, 101235).

## Animals
C57BL/6 J mice (6–8 weeks, female) and BALB/c Slc-nu mice (6–8 weeks, female) were purchased from Japan SLC, Inc. C.B-17/Icr-scid/scidJcl mice (6–8 weeks, male) with surgically induced ischemic stroke were purchased from Clea Japan, Inc. All animal husbandry and experimental procedures were approved by the Animal Care Use and Review Committee of Kyoto University. We have complied with all relevant ethical regulations for animal use. For ex vivo and in vivo studies, C57BL/6 mice were used unless otherwise indicated. BALB/c mice were only used for tumor xenograft experiments. C.B-17 mice were only used for brain cell isolation.

## Cell cultures
K562 (Riken BRC, RCB0027), HL60S (JCRB cell bank, JCRB0163) and HCC1806 (ATCC, CRL-2335) cells were grown in IMDM (Wako, 098-06465) containing 10% fetal bovine serum (FBS; Nichirei, 174012) and 1× penicillin-streptomycin solution (P/S; Wako, 168-23191) (hereafter GM). 3T3-L1 MBX (ATCC, CRL-3242) cells were cultured in DMEM (Wako, 043-30085) containing 10% FBS and 1× P/S. K562 cells were used in all assays. mCherry-expressing K562 cells were generated by infecting K562 cells with mCherry-expressing lentivirus[16], and only used for tumor xenografting. To induce adipocyte differentiation, 3T3-L1 MBX cells were seeded on 35 mm glass-base dish (IWAKI, 3971-035), and cultured for 48–72 hours to reach confluency. Cells were cultured in differentiation medium I (DMEM, 10% FBS, 1× P/S, 0.5 mM IBMX, 1 μg/mL insulin, 0.25 μM dexamethasone, and 2 μM rosiglitazone) for 48 hours, differentiation medium II (DMEM, 10% FBS, 1× P/S, and 1 μg/mL insulin) for 48 hours, and 10% FBS-containing DMEM for 48 hours.

## Azido-sugar uptake in living cells and post-click labeling
Prior to azido-sugar incorporation, cell density was adjusted. For suspension cell lines or primary cells, density was adjusted to $1 \times 10^6$ cells/mL. For adherent cell lines, $3 \times 10^5$ HCC1806 cells were seeded on a 35 mm glass bottom dish, and cultured for 20–24 h. Differentiated 3T3-L1 cells were prepared as described above. Cells were then washed with glucose-free IMDM [Gmep, custom-made: glucose was removed from the original IMDM (Wako, 098-06465)] twice, and incubated in glucose-free IMDM containing 10 mM azido-sugar at 25 °C for 60 mins, unless otherwise indicated. For overnight incubation in viability assays, cells were cultured in glucose-free IMDM supplemented with 20 μg/mL insulin, 110 μg/mL apo-transferrin, 13.4 ng/mL sodium selenite, 1 μg/mL L-ascorbic acid-2-phosphate at 37 °C for 20–24 h. After azido-sugar incorporation, cells were washed with 4% FBS/IMDM, and subsequently incubated with 100 nM BDP-DBCO diluted in 4% FBS/IMDM at 25 °C for 15 mins. Then, cells were washed with GM twice, and resuspended with GM for flow

cytometry or microscopy. To examine the effect of GLUT1 inhibitors (cytochalasin B and WZB-117) and competitive inhibitors (D-glucose, L-glucose, and 2DG), cells were washed with glucose-free IMDM twice, then pre-incubated with 1- or 10-μM GLUT1 inhibitors or 10- or 50-mM competitive inhibitors for 1 hr at 25 °C, followed by addition of 10 mM 6AzGal and labeling as described above. To examine the efflux of 6AzGal, cells treated with 6AzGal as described above were incubated in glucose-free IMDM, IMDM, or IMDM with 10 μM cytochalasin B at 25 °C for up to 120 mins, and labeled as described above.

## Flow cytometry
Cells were filtered and transferred into a 5 ml tube with cell strainer (35 μm pore size). The flow cytometry was performed using Sony Cell Sorter MA900, equipped with four excitation lasers (488, 405, 561, and 638 nm) and 12-color channels. All four lasers and filters with emission BP 525/50, 785/60, 450/50, 665/30, and 720/60 were used in this study. For optimal data acquisition, 100 μm sorting chips and following instrument settings were used for each cell types: K562: FSC threshold value: 5%; Sensor gain: FSC: 3, BSC: 33.5%, 525/50: 28.5%, and 665/30: 36.5%. HL60S: FSC threshold value: 5%; Sensor gain: FSC: 5, BSC: 36.0%, 525/50: 37.0%. Cells derived from spleen, thymus, blood, bone marrow, and brain tissues: FSC threshold value: 17%; Sensor gain: FSC: 11, BSC: 43.0%, 525/50: 43.5%, 785/60: 48.5%, 450/50: 45.0%, 665/30: 45.0%, and 720/60: 48.0%. For each sample, at least 30,000 events were analyzed. The data acquisition, analysis, and image preparation were carried out using the instrument software MA900 Cell Sorter Software (Sony). To conduct multi-color analysis with BDP-DBCO, fluorophores with a relative fluorescence intensity lower than 1% in the FL1 channel (BP 525/50) were used (Fig. S3).

## Confocal microscopy
To observe 6AzGal distribution in live K562 cell, 50 μL of cells labeled as described above was deposited on a 35 mm glass-base dish, allowed to settle on the bottom of the dish for 5 mins. HCC1806 and differentiated 3T3-L1 MBX cells cultured in a 35 mm glass-base dish were labeled as described above. Microscopy was performed using a Zeiss LSM800 confocal microscope with a Zeiss Plan-Apochromat ×63/1.40 oil objective. The data acquisition, analysis, and image preparation were carried out using the instrument software ZEN (ZEISS).

## 2DG and 2NBDG uptake assays
For measuring 2DG uptake, Glucose Uptake-Glo Assay (Promega, J1341) was performed as manufacturer's instruction. Briefly, cells were rinsed with glucose-free IMDM twice, then 50 μL of $1 \times 10^5$ K562 cells was seeded per well of a white 96-well plate. Cells were then incubated with 1 mM 2DG at 25 °C for 1 hour, and followed by subsequently adding stop buffer, neutralization buffer, and detection buffer. Luminescence was measured with a plate reader (TECAN, Infinite® 200 PRO). For measuring 2NBDG uptake, K562 cells were pre-washed with glucose-free IMDM, incubated with 300 μM 2NBDG at 37 °C for 30 mins[14], then washed with GM twice, and subjected to flow cytometric analysis. To examine the effect of GLUT1 inhibitors on 2DG/2NBDG uptake, cells were pre-treated with GLUT1 inhibitor as described above, prior to the 2DG/2NBDG incubation. For 2DG uptake assay on purified splenic T and B cells, cells were first isolated with EasySep™ Mouse T cell Isolation kit (StemCell™, 19851) and EasySep™ Mouse B cell Isolation kit (StemCell™, 19854) as manufacturer's instruction respectively, followed by Glucose Uptake-Glo Assay.

## Cell viability assays
After K562 cells were treated with 6AzGal and labeled with BDP-DBCO as described above, following cell assays were conducted. For trypan blue assay, 50 μL of cell suspension was mixed with equal volume of 0.4% (v/v) trypan blue solution, and then cell numbers were counted with Automated Cell Counter (ThermoFisher). For propidium iodide (PI)/annexin V-Alexa Fluor 488 apoptosis detection assay (ThermoFisher, A13201), 200 μL of cell

suspension was transferred to 1.5 mL tube, mixed with 1 µL each stock solution of PI and annexin V, incubated at 25 °C for 15 mins, then analyzed with flow cytometry. For Cell Counting Kit-8 (CCK8) assay (Wako, 341-07761), 100 µL of cell suspension was inoculated on a 96-well plate per well. 10 µL of CCK8 stock solution was added into each well and incubated at 37 °C for 1 hour. The absorbance at 450 nm was measured with a plate reader.

### Ex vivo 6AzGal uptake assays

For analysis of splenocytes, mice were euthanized by cervical dislocation, and spleen was harvested according to the previous protocol[31]. The tissue was placed on 70 µm cell strainer inserted on top of the 50 ml conical tube, moistened with 2% FBS/PBS, then dilacerated with the plunger of a 3 ml syringe, followed by centrifugation at $450 \times g$ for 7 mins to pellet the splenic cells. Cells were resuspended with 1 mL of VersaLyse solution (Beckman, A09777), and incubated at 25 °C for 15 mins. After the erythrocyte lysing step, 5 mL of 2% FBS/PBS was added and centrifuged at $450 \times g$, 7 mins, then resuspended with glucose-free IMDM or GM to $2–5 \times 10^6$ cells/mL. 6AzGal incorporation was performed as described above, followed by immunolabelling with fluorescence-conjugated antibodies in the presence of FVD780 in GM at 25 °C for 15 mins, washed with 4% FBS/IMDM twice, then labeled with BDP-DBCO and subjected to flow cytometric analysis as described above. For analysis of CD3/CD28-stimulated T cells, T cells were purified from a splenic suspension with EasySep™ Mouse T cell Isolation kit as manufacturer's instruction. Purified splenic T cells were resuspended in GM at a cell density of $1 \times 10^6$ cells/mL, then $2 \times 10^6$ cells (2 mL) were dispatched in each well of a six-well plate. To activate T cells, 20 µL of Dynabeads Mouse T-activator CD3/CD28 (ThermoFisher, 11456D) was added into the well, and cultured at 37 °C for 48 hours. Stimulated and non-stimulated cells were then collected, followed by 6AzGal incorporation, immunolabeling, BDP-DBCO labeling as described above, and finally analyzed with flow cytometry.

### In vivo 6AzGal uptake assays

Twenty mg/kg of 6AzGal was administered intraperitoneally (for spleen, thymus, blood, and bone marrow) or retro-orbitally (brain) into fasted mice and circulated for 30 mins, or otherwise indicated. After the mice were sacrificed, tissues of the interest were collected and processed as follows. Spleen and thymus were subjected to preparation of single cell suspension as described above in the ex vivo 6AzGal uptake assays in splenocytes. Whole blood was collected by cardiac puncture in heparin tubes and treated with VersaLyse solution to lyse erythrocytes as the manufacturer's instruction. White blood cells were obtained by centrifugal separation. Bone marrow-derived cells were collected by flushing the femur and tibia bone marrow with PBS according to the previous protocol[32]. For brain cell purification, isolation of myelin-free brain cells was performed according to the previous protocol[33] with modification. Briefly, after removal of olfactory bulbs, midbrain, cerebellum and hindbrain, remaining forebrain was minced into smaller pieces with a surgical blade. Tissues were treated with 2 mg/mL collagenase (Wako, 038-22361), 28 U/mL DNase I (NipponGene, 314-08071), 5% FBS, 10 µM HEPES (Wako, 345-06681) in 1× PBS ($Mg^{2+}/Ca^{2+}$-free) at 37 °C for 30 mins, then dissociated with 1000 µL pipet tip, and filtered through 70 µm cell strainer to remove debris and undissociated cell clusters, followed by 30%/70% Percoll gradient to remove myelin and red blood cells. Purified cells were then immunolabelled with fluorescent-tagged antibodies for 15 mins at 25 °C in GM, followed by BDP-DBCO labeling and flowcytometric analysis. For accurate measurements, 10 µM cytochalasin B was constantly supplied to prevent efflux of 6AzGal. To determine the effect of lipopolysaccharide (LPS; Wako, 125-05181) on 6AzGal uptake in vivo, 50 mg/kg of LPS was intraperitoneally administrated, and allowed to be absorbed and circulated for 4 hours prior to 6AzGal administration. To determine the inhibitory effect of D-glucose on 6AzGal uptake in vivo, 8 mg/kg of 6AzGal were intraperitoneally injected with or without 12 mg/kg of D-glucose.

### Subcutaneous tumor xenograft

Stable mCherry-expressing K562 cells were harvested, washed twice in PBS and resuspended in IMDM at density of $1 \times 10^7$ cells/mL. One million cells were inoculated subcutaneously into the dorsal side of the nude mice. Xenografts were then grown for 2–3 weeks. After tumor became visibly obvious (1.5 cm × 1 cm × 0.5 cm at least), xenografted mice were injected i.p. with 10 mg/kg of WZB-117 (ref. 14) 1 hour before administration of 6AzGal. Xenograft tumors were harvested and placed on 70 µm cell strainer inserted on top of the 50 ml conical tube, moistened with 2% FBS/PBS, then dilacerated with the plunger of a 3 ml syringe, followed by centrifugation at $450 \times g$ for 7 mins to pellet the splenic cells. Cells were resuspended with 1 mL VersaLyse Buffer, and incubated at 25 °C for 15 mins. After the erythrocyte lysing step, 5 mL of 2% FBS/PBS was added and centrifuged at $450 \times g$, 7 mins, then resuspended with glucose-free IMDM or GM to $2–5 \times 10^6$ cells/mL. Cells were then labeled and analyzed as described above.

### Statistics and reproducibility

All data were represented as mean ± SEM. Statistical analyses were performed using a two-way unpaired $t$ test. Sample sizes were included in figures or legends. For flow cytometric assays, the sample sizes indicate numbers of independent cell culture (Figs. 3d, e, S2c, h–l, n, o) and individual mice (Figs. 4b, e–h and S4d, g, S5b, c, S6d). For confocal imaging, the sample sizes indicate numbers of individual cells (Figs. 3f and S2m). All the experiments were repeated at least three times. Linear fitting and the corresponding $R^2$ values (Figs. 4c and S2i, j, S5b) were obtained using Microsoft Excel (detailed data and calculation were present in Source Data file).

### Reporting summary

Further information on research design is available in the Nature Portfolio Reporting Summary linked to this article.

### Data availability

All data that support the findings are available within the manuscript and the Supplementary Information. Numerical source data for the graphs in the manuscript are available in Supplementary Data 1. Other data supporting this study are available from the corresponding author on reasonable requests.

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

## Acknowledgements

We thank Mitchell Arico from Edanz (https://jp.edanz.com/ac) for editing a draft of this manuscript. This work was supported by a Grant-in-Aid for Specially Promoted Research (JSPS KAKENHI Grant 23H05405) and the Japan Science and Technology Agency (JST) ERATO Grant JPMJER1802 to I.H., and a Grant-in-Aid for Early-Career Scientists (22K15060), a Grant-in-Aid for Transformative Research Areas (23H03856), JST PRESTO Grant JPMJPR20EA, the Terumo Life Science Foundation and the Japan Health Foundation to M.T.

## Author contributions

M.T. and I.H. conceived the project and designed the experiments. M.T. and N.T. performed the experiments and data analysis. M.T., N.T. and I.H. wrote the manuscript with input from all authors.

## Competing interests

The authors declare no competing interests.
