## [Peer Review File · Communications Biology]

Reviewers' comments:

Reviewer #1 (Remarks to the Author):

The manuscript entitled "Post-click labeling enables highly accurate single cell analyses of glucose uptake ex vivo and in vivo" presents a detailed story about the development of a small-sized azide-tagged sugar, which acts as a more selective tracer for GLUT transport. Their evidence, supported by both in vitro and in vivo experiments, convincingly demonstrates GLUT-dependent cellular uptake of azido-sugars. The experiments conducted in this study were robust and appropriate for the objectives of the study. The concept introduced in this paper, although straightforward, is novel in the context of glucose transportation probes. Therefore, we recommend the publication of this paper in *Communications Biology*, after minor revision based on the following questions and minor points:

- GLUTs are well known transporters for diverse sugars, including glucose, galactose, and even sometimes fructose. However, the transporting availability of each monosaccharide analog varies with each GLUT homolog. Authors utilized the galactose probe rather than glucose probe due to the data of Fig S2. Though authors committed several experiments to verify the uptake of 6AzGal is dependent on GLUT, they should verify two more points: 1. From where the discrepancy between the uptake tendency of each azido sugar analogs (shown in Fig S2C) originates; 2. Which GLUT homolog is most responsible for the activity of 6AzGal.
- In their previous works (ref. 15, 16), the authors have shown that BDP-DBCO localizes in the ER/Golgi. This property was harnessed for ER/Golgi-targeting using an azido-stained organelle membrane, a concept bearing the resemblance to this study. In this manuscript, the authors haven't addressed this. Based on the morphology evident in Fig. 1e, we hypothesized that BDP-DBCO might still localize in the ER. Could the authors provide commentary or perhaps present the co-localization experiment data related to the BDP-DBCO-Azido-sugar signals? If there were indeed ER accumulation of BDP-DBCO, are there potential implications? While we presume that there might not be any issues, especially concerning the FACS experiment (a main objective of this study), a brief discussion would be valuable for a comprehensive understanding of the probe system.
- The choice of BDP-DBCO as the preferred fluorophore raises a few questions. In previous studies (ref. 15, 16), other organelle-targeting fluorophores were utilized by the authors. Why was BDP-DBCO, which targets the ER, deemed the most suitable for this study, especially when various other fluorophores could be available in their lab? Could the authors elaborate on the results or potential outcomes of using a mitochondria-targeting fluorophore-DBCO (as employed in ref. 16) with the azido-sugar system? If there's minimal or no dependency on the fluorophore, it would present a significant advantage in terms of emission-color variations. Moreover, utilizing a fluorophore system that's evenly distributed within the cytoplasm might enhance the efficacy of this tracer system. We suggest that the authors either present a comparison of BDP-DBCO with other fluorophore-DBCOs or provide data and discussions to support the superiority of BDP-DBCO among available fluorophore-DBCO options.
- Strain promoted click reaction between DBCO & Azide is well known chemistry. The use of this reaction should ensure low backgrounds as mentioned by authors. However, Fig 1e, there are quite strong background with no azide probe. Interestingly Fig S2d shows clean imaging data without this kind of backgrounds. Authors should explain from where this high background originates.

Followings are the minor points, and some recommendations to increase visibility.

- The current figures, especially Figure 1, are overloaded with sub-figures. For enhanced clarity, it would be advisable to split the primary figures. In Fig 1f, authors claimed that L-glucose showed only marginal effect on 6AzGal uptake. However, effective concentrations for each competitor vary. Authors are recommended to provide dose dependency with wider range of concentrations for each competitor in supporting data. In Fig 1f, g, and h, it is recommended to indicate p-values
- We recommend to include the exact structure of 6AzGal (the best one) in the main figures, not only in supplementary figures.
- Quantitively in line 76 is recommended to be changed to Quantitatively. Though quantitatively can be an alternative to quantitatively, quantitatively is more adequate for the edited writings

Reviewer #2 (Remarks to the Author):

Title: Post-click labeling enables highly accurate single cell analyses of glucose uptake ex vivo and in vivo

The authors are correct in that the single cell field is desperate for a clean, clear-cut glucose uptake assay.

However, this assay as they present it in this current form, is not it.

There are several issues within the paper that underlie this reasoning:

The decision to do the uptake assays at 10mM for 1 hr is concerning. It is likely that this is oversaturated – in fact the data in fig 2Sg and h confirm that uptake is saturated/plateaued by 30 mins (at 10mM) in K562 cells. For this assay to be a useful point of reference for glucose uptake, the amount of substrate taken up by the cells should be a true reflection of the cells capacity to take up nutrients/glucose. The assay cannot reflect this accurately if it is already saturated.

There are significant concerns about the efficiency of transport of the 6AzGal substrate - indeed galactose is NOT a transported substrate by many glucose transporters, and the azide modification may well alter it's transport capacity. The authors have not performed any titrated competition assays to establish how their substrate performs against uptake of bonafide glucose transporter substrates, whether they have comparable Km or not; whether it is preferential for one GLUT over another...

Back to the uptake assay itself as presented; the 10mM uptake substrate (6AzGal) is detected by 100nM BDP-DBCO. This is 100'000 fold LESS than the uptake substrate. There are serious concerns with this, a couple of which are : are the authors looking at limitations of probe uptake or limitations in the ability to effectively label with the DBCO?

The BDP-DBCO dye used to “click” the 6AzGal substrate clearly has different background staining levels on different cell populations. Indeed the strained alkyne type reagents like the DBCO are known to have

some nonspecific reactivities inside cells, and it is not unreasonable for the level of background reactivity to differ between cell types. Whilst it is understandable that the authors aim to correct this by calculating the fold change of no azide to 6AzGal MFI, this is inappropriate as a measure of substrate uptake. It normalises to a non-fixed and of itself intrinsically variable parameter (variations are clear in fig S5, the 6AzGal “top” staining” doesn’t shift from 10e4, whereas the control “non azide” is what varies, and thereby “changes” the uptake measurement).

How does the uptake assay perform with copper mediated click reaction, what is the signal/noise ratio looking like here? (Different non-specificities inherent in this system too!)

The authors use fold change in comparison to no azido-sugar in some graphs, but in others (eg. Fig 1f&g, Fig S2j) a normalization to maximum uptake is used instead. For these graphs specifically, the level of background labeling (so the same no azido-sugar control) should be shown as well. As the data is presented now, it is impossible to judge whether the observed remaining signal in these experiments is within the limit of detection of the assay.

When considering the in vivo high thymocyte staining as shown in fig2. Thymocytes, as a bulk population, do not take up high levels of glucose (Immunometabolism 2020 Aug 17;2(4):e200029. doi: 10.20900/immunometab20200029. eCollection 2020.), and yet the thymocytes stain very brightly above “background” for 6AzGal presence/uptake.

It appears that the 6AzGal uptake assay is acting in a similar fashion to 2NBDG in this instance... therefore how is this to be a trusted uptake assay??

~ Perhaps consistent use of transporter inhibitors is required as a vital control in each assay?

In working up this assay, the authors should be commended on their use of transporter controls, and comparisons with valid glucose uptake assays, as well as comparisons with existing flawed assays which they are trying to better. However, it is clear from the data presented that much more is needed to be known about the transport parameters of 6AzGal. Furthermore, the output or “uptake measurement” CANNOT be governed by the variability in the labelling of the detection reagent.

The authors need to nail down these issues in order for this assay to be any better than the flawed assays already in existence. Specifically – demonstrably linear uptake parameters which conform to physiological transporter parameters AND the output should not be reliant on background correction which is highly variable in the absence of the uptake substrate.

Point-by-Point Responses to Reviewers

The followings are our responses (highlighted in BLUE) to all of referees' comments.

Reviewers' comments:

Reviewer #1 (Remarks to the Author):

The manuscript entitled "Post-click labeling enables highly accurate single cell analyses of glucose uptake ex vivo and in vivo" presents a detailed story about the development of a small-sized azide-tagged sugar, which acts as a more selective tracer for GLUT transport. Their evidence, supported by both in vitro and in vivo experiments, convincingly demonstrates GLUT-dependent cellular uptake of azido-sugars. The experiments conducted in this study were robust and appropriate for the objectives of the study. The concept introduced in this paper, although straightforward, is novel in the context of glucose transportation probes. Therefore, we recommend the publication of this paper in *Communications Biology*, after minor revision based on the following questions and minor points:

We thank the reviewer for the positive assessment of this work. We addressed each of the questions and minor points in the following paragraphs.

- GLUTs are well known transporters for diverse sugars, including glucose, galactose, and even sometimes fructose. However, the transporting availability of each monosaccharide analog varies with each GLUT homolog. Authors utilized the galactose probe rather than glucose probe due to the data of Fig S2. Though authors committed several experiments to verify the uptake of 6AzGal is dependent on GLUT, they should verify two more points: 1. From where the discrepancy between the uptake tendency of each azido sugar analogs (shown in Fig S2C) originates;

Response #1-1: Bar graphs in Fig. S2c show the levels of fluorescently labeled azido sugars in cells, which reflect both the uptake capability and the subsequent click reaction efficiency. Among the tested azido sugar analogs, we found that 6AzGal, 6AzGlc and 6AzFuc possessing a hydrophobic azide group at the C-6 position showed higher fluorescence signals than the other isomers that have an azide at the C-1, C-2, C-3 or C-4 position. This observation agrees with previous studies indicating that substitution of the C-6 hydroxyl group with a hydrophobic group enhanced the recognition of the hexose analogs (especially galactose) by GLUTs (*Sensors* 12, 5005 (2012); *Biochem. J.* 131, 211 (1973); *Biochem. Biophys. Res. Commun.* 480, 341 (2016)). Besides, the previous studies reported that the reaction efficiency between azide and DBCO is influenced by steric hindrance (*Chemistry* 25, 754 (2019); *Curr. Opin. Chem. Biol.* 60, 79 (2021)). It is well-known that the C-6 position is less sterically crowded than the other position such as C-1,2,3,4, which let us reasonably expect that the azides in 6AzGal, 6AzGlc and 6AzFuc are more reactive with BDP-DBCO than the other azides. These two effects may be combined (C-6 hydrophobic substitution and steric hindrance), which should contribute to the difference of labeling output among azido sugar analogs (Fig S2c). These comments are added

in the discussion part.

2. Which GLUT homolog is most responsible for the activity of 6AzGal.

Response #1-2: In this study, we selected HCC1806 cells in which GLUT1 is reported to be the major pathway for glucose uptake (*Nat. Commun.* 11, 4205 (2020)). Indeed, our inhibition assay on HCC1806 cells (Fig. 3f) indicates that 6AzGal is mainly transported by GLUT1. Also, our stimulation assay (activated CD4⁺ T cells showed the increased GLUT1 activity (*Cell Metab.* 20, 61 (2014))) revealed an increase in 6AzGal labeling (Fig. 4c), confirming GLUT1-mediated 6AzGal uptake. These observations are consistent with the fact that galactose acts as a substrate of GLUT1 (*Biochemistry* 31, 10414 (1992); *Mol. Aspects Med.* 34, 121 (2013)). Thus, it is reasonably concluded that GLUT1 is the major contributor for 6AzGal uptake. These comments are added in the discussion part.

- In their previous works (ref. 15, 16), the authors have shown that BDP-DBCO localizes in the ER/Golgi. This property was harnessed for ER/Golgi-targeting using an azido-stained organelle membrane, a concept bearing the resemblance to this study. In this manuscript, the authors haven't addressed this. Based on the morphology evident in Fig. 1e (Fig. 3c in the revised manuscript), we hypothesized that BDP-DBCO might still localize in the ER. Could the authors provide commentary or perhaps present the co-localization experiment data related to the BDP-DBCO-Azido-sugar signals? If there were indeed ER accumulation of BDP-DBCO, are there potential implications? While we presume that there might not be any issues, especially concerning the FACS experiment (a main objective of this study), a brief discussion would be valuable for a comprehensive understanding of the probe system.

Response #1-3: As the reviewer pointed out, most of the BDP-DBCO-labeled 6AzGal was localized in the ER/Golgi (Fig. 3c). This result together with our previous imaging approach (Tamura, *Nat. Chem. Biol.* 2020) raised an interesting possibility of tracing azido-sugar dynamics in live cells. It should be noted, however, that 6AzGal undergoes drastic changes in its physicochemical properties upon BDP-DBCO labeling (MW: from 205.2 to 755.6, ClogP: from -0.78 to 2.5), which might bias the true distribution of 6AzGal as a small hydrophilic (water-soluble) sugar. This is different from the case of phospholipid labeling. Although other DBCO reagents such as mitochondria-targeting Cy3-DBCO might have a potential to expand organelle-selective azido-sugar labeling, the scope of this work was focused on validating flow cytometric 6AzGal uptake assay rather than the imaging applications. These comments are added in the discussion section.

- The choice of BDP-DBCO as the preferred fluorophore raises a few questions. In previous studies (ref. 15, 16), other organelle-targeting fluorophores were utilized by the authors. Why was BDP-DBCO, which targets the ER, deemed the most suitable for this study, especially when various other

fluorophores could be available in their lab? Could the authors elaborate on the results or potential outcomes of using a mitochondria-targeting fluorophore-DBCO (as employed in ref. 16) with the azido-sugar system? If there's minimal or no dependency on the fluorophore, it would present a significant advantage in terms of emission-color variations. Moreover, utilizing a fluorophore system that's evenly distributed within the cytoplasm might enhance the efficacy of this tracer system. We suggest that the authors either present a comparison of BDP-DBCO with other fluorophore-DBCOs or provide data and discussions to support the superiority of BDP-DBCO among available fluorophore-DBCO options.

Response #1-4: The reason why we selected BDP-DBCO in this study was that BDP-DBCO previously showed the highest sensitivity in intracellular fluorescence labeling of azide-tagged phosphatidylcholine (Tsuchiya, *Cell Metab.* 2023). Compared to the other DBCO reagents (e.g. mitochondria-targeting Cy3-DBCO), BDP-DBCO can be more efficiently concentrated in cells and readily removed (washed out) by medium exchange. We therefore anticipated that BDP-DBCO transported inside cells could react with azido sugars to obtain higher labeling yield relative to others. On the other hand, mitochondria-targeting DBCOs are the less effective in the sensitivity and also need more complicated steps for selective labeling and reducing the background signals (Tsuchiya, *STAR Protoc.* 2023). Because we have established the sufficiently useful flow cytometric 6AzGal uptake assay (the main objective of this study) using BDP-DBCO, we did not test the mitochondria-targeting DBCOs in this study. As the reviewer recommended, the emission-color variation may be crucial for some experiments. In order to reply to the request, we newly tested the blue-emitting BDP-DBCO variant (8AB-DBCO) and showed the different color 6AzGal labeling (newly added in Figs. S2d and S2e). But, due to the lower brightness of 8AB-DBCO than BDP-DBCO (*Chemistry* 17, 7261 (2011)), we used BDP-DBCO in this study. In term of a fluorophore system that's evenly distributed within the cytoplasm, we unfortunately could not develop such an interesting probe. These comments are added the discussion part.

- Strain promoted click reaction between DBCO & Azide is well known chemistry. The use of this reaction should ensure low backgrounds as mentioned by authors. However, Fig 1e (Fig. 3c in the revised manuscript), there are quite strong background with no azide probe. Interestingly Fig S2d (Fig. S2f in the revised manuscript) shows clean imaging data without this kind of backgrounds. Authors should explain from where this high background originates.

Response #1-5: We are concerned that the reviewer might misunderstand these imaging data. Images in Fig. 3c showed the adherent HCC1806 cells, while Fig. S2f showed floating K562 cells. These data were not obtained in the same way. These imaging data just indicated the representative intracellular distribution of BDP-DBCO-labeled 6AzGal in a qualitative manner, and thus the image contrast was tuned for increasing visibility in a printed paper. Our quantitative analysis of BDP-DBCO fluorescence

showed ~10-fold increase in the presence of 6AzGal in the both cells, revealing the sufficient low backgrounds (Figs. 3b and 3f).

Followings are the minor points, and some recommendations to increase visibility.

- The current figures, especially Figure 1 (Figs. 1, 2, and 3 in the revised manuscript), are overloaded with sub-figures. For enhanced clarity, it would be advisable to split the primary figures. In Fig 1f (Fig. 3d in the revised manuscript), authors claimed that L-glucose showed only marginal effect on 6AzGal uptake. However, effective concentrations for each competitor vary. Authors are recommended to provide dose dependency with wider range of concentrations for each competitor in supporting data. In Fig 1f, g, and h (Figs. 3d, 3e, and 3f in the revised manuscript), it is recommended to indicate p-values

Response #1-6: According to the reviewer's recommendation, we split the previous Figure 1 to the current Figures 1, 2, and 3. We also added *p* values in Figs. 3d, 3e, and 3f. The statistical analysis (Fig. 3d) showed the significant differences (50 mM L-glucose vs 50 mM D-glucose, and 50 mM L-glucose vs 50 mM 2DG), confirming that the inhibitory effect of L-glucose on 6AzGal uptake was much weaker than D-glucose and 2DG. In the previous studies of sugar probes monitoring GLUT-dependent glucose uptake (*Nat. Commun.* 13, 5974 (2022); *Angew. Chem.* 54, 9821 (2015)), competitive assays were performed by using three samples of high-concentration D-glucose, low-concentration D-glucose, and high-concentration L-glucose, which are the same conditions as our study. Based on these previous studies, we are convinced that our current data set (Fig. 3d) adequately supports the minimal effect of L-glucose on 6AzGal uptake.

- We recommend to include the exact structure of 6AzGal (the best one) in the main figures, not only in supplementary figures.

Response #1-7: The structure of 6AzGal was presented in Fig. 3a.

- Quantitively in line 76 (line 87 in the revised manuscript) is recommended to be changed to Quantitatively. Though quantitatively can be an alternative to quantitatively, quantitatively is more adequate for the edited writings

Response #1-8: We changed the word to "quantitatively".

Reviewer #2 (Remarks to the Author):

Title: Post-click labeling enables highly accurate single cell analyses of glucose uptake ex vivo and in vivo

The authors are correct in that the single cell field is desperate for a clean, clear-cut glucose uptake assay.

However, this assay as they present it in this current form, is not it.

There are several issues within the paper that underlie this reasoning:

We thank the reviewer for examination of our manuscript.

The decision to do the uptake assays at 10mM for 1 hr is concerning. It is likely that this is oversaturated – in fact the data in fig 2Sg and h (Figs. S2i and S2j in the revised manuscript) confirm that uptake is saturated/plateaued by 30 mins (at 10mM) in K562 cells. For this assay to be a useful point of reference for glucose uptake, the amount of substrate taken up by the cells should be a true reflection of the cells capacity to take up nutrients/glucose. The assay cannot reflect this accurately if it is already saturated.

Response #2-1: We are concerned that the reviewer may misunderstand our work. Figure S2i shows the concentration-dependent fluorescence increase in the range from 0 to 20 mM of 6AzGal for 1 hour incubation. In these 6AzGal concentration range, good linearity of the fluorescence output was observed in both raw MFIs ($R^2 = 0.95$) and MFI fold changes ($R^2 = 0.94$). Especially, at the points of 5, 10, and 20 mM 6AzGal, MFIs were clearly increased in a concentration-dependent manner, clearly indicating that no evidence of oversaturated signals in our assay conditions (also please see **Response #2-9**). If 6AzGal uptake is oversaturated at 10 mM for 1 hour as the reviewer mentioned, the inhibitory effects should not be detected in our assay. However, robust fluorescence decreases were actually detected in all the inhibitory experiments (Figs. 3d, 3e, 3f, S2k, and S2l). We believe that all of these results can exclude possibility of oversaturation in our assay (10 mM for 1 hour). In addition, we optimized our assay conditions by taking into consideration the experimental conditions of the previously validated study (*Angew. Chem.* 54, 9821 (2015), Citations: 137). In the previous study, 3-OPG uptake assay was performed at 25 mM for 4 hours, clearly showing the decreased signals by GLUT inhibition despite the fact that 3-OPG uptake reaches a plateau within 30 mins. Given these results and reference, we could not find any poor accuracy in our assay.

There are significant concerns about the efficiency of transport of the 6AzGal substrate - indeed galactose is NOT a transported substrate by many glucose transporters, and the azide modification may well alter it's transport capacity. The authors have not performed any titrated competition assays to establish how their substrate performs against uptake of bonafide glucose transporter substrates, whether they have comparable K_m or not; whether it is preferential for one GLUT over another...

Response #2-2: We are again afraid that the reviewer misunderstands the substrate specificity in GLUT transporters. It is well established that galactose acts as a substrate of GLUTs (*Biochemistry* 31, 10414 (1992) citation: 261; *Mol. Aspects Med.* 34, 121 (2013) citation: 1303), as this is also recognized

by the Referee #1 (please see the second paragraph in the Referee #1's comments). As we described mechanistic discussion of 6AzGal uptake and labeling in **Response #1-1**, previous papers reported substitution of the C-6 hydroxyl group with a hydrophobic group would enhance the GLUT recognition (*Sensors* 12, 5005 (2012); *Biochem. J.* 131, 211 (1973); *Biochem. Biophys. Res. Commun.* 480, 341 (2016)). In addition, 6AzGal has the primary azide at the less crowded C-6 position which can accelerates the click labeling reaction with BDP-DBCO (*Chemistry* 25, 754 (2019); *Curr. Opin. Chem. Biol.* 60, 79 (2021)). These combined effects contributed to effective detection of 6AzGal taken up by cells with the sufficient sensitivity (Fig. S2c). To confirm the GLUT-mediated 6AzGal uptake, we performed the competitive experiments using three samples of high-concentration D-glucose, low-concentration D-glucose, and high-concentration L-glucose, which are the same conditions as the previous studies of GLUT-mediated glucose-uptake probes (*Nat. Commun.* 13, 5974 (2022); *Angew. Chem.* 54, 9821 (2015)). We found 6AzGal uptake was decreased by D-glucose and 2DG in a concentration-dependent manner, while high-concentration L-glucose showed only a subtle change (Fig. 3d). These results well agree with those previous studies (*Nat. Commun.* 13, 5974 (2022); *Angew. Chem.* 54, 9821 (2015)). Moreover, our inhibition assay was conducted using HCC1806 cells in which GLUT1 is the major route for glucose uptake (*Nat. Commun.* 11, 4205 (2020)), indicating that 6AzGal is transported by GLUT1. This result is also quite consistent with the fact that galactose is a substrate of GLUT1 (*Biochemistry* 31, 10414 (1992); *Mol. Aspects Med.* 34, 121 (2013)).

Regarding K_m values in 6AzGal uptake, we agree with the reviewer that the K_m determination will strengthen reliability of our work. But we could not evaluate the amount of 6AzGal taken up by cells, because quantification of low-molecular weight 6AzGal in cell lysate was technically difficult in mass spectrometric analysis. Generally, K_m determination is recognized as a hard task due to technical requirements to quantify absolute abundances of target molecules. Therefore, K_m values have been seldom determined in the many previous reports of glucose-uptake probes (*Nat. Commun.* 13, 5974 (2022); *Anal. Chem.* 94, 8293 (2022); *Chem. Commun.* 56, 1070 (2020); *Nat. Methods* 16, 526 (2019); *Bioconjug. Chem.* 29, 3394 (2018); *Biochem. Biophys. Res. Commun.* 480, 341 (2016); *Biochem. Biophys. Res. Commun.* 474, 240 (2016); *Org. Biomol. Chem.* 9, 4760 (2011); *Angew. Chem.* 48, 8027 (2009) etc.).

Back to the uptake assay itself as presented; the 10mM uptake substrate (6AzGal) is detected by 100nM BDP-DBCO. This is 100'000 fold LESS than the uptake substrate. There are serious concerns with this, a couple of which are : are the authors looking at limitations of probe uptake or limitations in the ability to effectively label with the DBCO?

Response #2-3: As we are concerned that the reviewer might incorrectly understand our labeling system, we would like to explain its basic background and mechanism in detail. Standard basal culture media contain ~25 mM glucose, and GLUTs transport glucose by its concentration gradient with the

low affinity (K_m in the mM range) (*Biochemistry* 31, 10414 (1992); *Mol. Aspects Med.* 34, 121 (2013)). In our assay, the linearity in the 6AzGal concentration was revealed up to 20 mM (Fig. S2i), which is almost same as the well-validated glucose-uptake probes, 2DG and 3-OPG (Glucose Uptake-Glo™ Assay Manual (Promega); *Angew. Chem.* 54, 9821 (2015)). According to the previous report (*Angew. Chem.* 54, 9821 (2015)), sugar analogs at 20 mM in extracellular medium can produce an intracellular concentration in the mM range. On the other hand, we previously demonstrated that BDP-DBCO (at 100 nM in extracellular medium) concentrates inside of cells to ~1 mM (due to the effective (spontaneous) uptake, in *Nat. Chem. Biol.* 16, 1361 (2020)). Eventually, relatively high concentration of 6AzGal and BDP-DBCO inside of live cells accelerate a spontaneous click reaction to produce BDP-DBCO-labeled 6AzGal (otherwise this Click reaction between azide and DBCO is not so rapid based on its moderate second-order rate constant) (Fig. 1). These comments are added in the discussion part.

The BDP-DBCO dye used to “click” the 6AzGal substrate clearly has different background staining levels on different cell populations. Indeed the strained alkyne type reagents like the DBCO are known to have some nonspecific reactivities inside cells, and it is not unreasonable for the level of background reactivity to differ between cell types. Whilst it is understandable that the authors aim to correct this by calculating the fold change of no azide to 6AzGal MFI, this is inappropriate as a measure of substrate uptake. It normalises to a non-fixed and of itself intrinsically variable parameter (variations are clear in fig S5, the 6AzGal “top” staining” doesn’t shift from $10e4$, whereas the control “non azide” is what varies, and thereby “changes” the uptake measurement).

Response #2-4: We agree with the reviewer comment that adequate controls are critical for comparison of fluorescence signals in different cell populations with different background levels. As we confirmed that treatment of cells with 6AzGal does not change cellular autofluorescence, we prepared 6AzGal (+) & BDP-DBCO (+)-treated cells (6AzGal sample) and 6AzGal (-) & BDP-DBCO (+)-treated cells (no azide control) for our quantitative analysis throughout this study. Although the reviewer might miss the details of our data, our validation study carefully takes into account the differences in the background fluorescence levels. We quantitatively characterized MFI fold changes only in cell populations with the same background level, and did not directly compare MFIs in cell populations with different background levels. Because the background fluorescence intensities in splenic T and B cells were confirmed to be at almost the same level, we compared MFI fold changes for quantitative analysis of 6AzGal uptake (Fig. 4b, S4b, S4c, S5d). On the other hand, in analyzing bone marrow CD45+ cells showing two populations with different background levels (Fig. 4d, and S5a), we just provided a qualitative description of the fluorescence increase upon BDP-DBCO labeling of 6AzGal. When more quantitative comparison of fluorescence changes among different cell populations are required, additional cell surface markers of interest (e.g. CD11b for myeloid cells) can

be used to narrow down cell kinds with the same background. Moreover, although MFI fold changes were mainly used in this study, an MFI difference between non-azide- and 6AzGal-treated cells can be alternatively used depending on the applications and research purposes.

How does the uptake assay perform with copper mediated click reaction, what is the signal/noise ratio looking like here? (Different non-specificities inherent in this system too!)

Response #2-5: Unfortunately, copper click chemistry is not applicable to live cell systems because of its Cu toxicity and the limited experimental conditions. Given that copper click reaction needs cell membrane-impermeable reagents (e.g. copper ions and chelators), copper-click labeling of azido molecules inside the cells requires cell fixation and membrane permeabilization processes. It is heavily concerned that azido sugars are eventually washed out from cells during the process, which makes it more challenging to accurately capture azido-sugar uptake using copper click chemistry. Therefore, we selected copper-free click chemistry in this study.

The authors use fold change in comparison to no azido-sugar in some graphs, but in others (eg. Fig 1f&g, Fig S2j (Figs. 3d, 3e, and S2l in the revised manuscript)) a normalization to maximum uptake is used instead. For these graphs specifically, the level of background labeling (so the same no azido-sugar control) should be shown as well. As the data is presented now, it is impossible to judge whether the observed remaining signal in these experiments is within the limit of detection of the assay.

Response #2-6: In these figures, we always include no azido-sugar control to determine the net fluorescence increase in labeling of 6AzGal with BDP-DBCO. For example, in Figs. 3d and 3e, raw MFIs in three samples (6AzGal with inhibitor, 6AzGal without inhibitor, and no azide) were used to calculate 6AzGal uptake levels (y axis in the bar graphs) as follows: $[\text{MFI}(6\text{AzGal with inhibitor}) - \text{MFI}(\text{no azide})] \div [\text{MFI}(6\text{AzGal without inhibitor}) - \text{MFI}(\text{no azide})] \times 100$. According to the reviewer's comment, we added representative histograms in Fig. 3d.

When considering the *in vivo* high thymocyte staining as shown in fig2 (Fig. 4 in the revised manuscript). Thymocytes, as a bulk population, do not take up high levels of glucose (Immunometabolism 2020 Aug 17;2(4):e200029. doi: 10.20900/immunometab20200029. eCollection 2020.), and yet the thymocytes stain very brightly above “background” for 6AzGal presence/uptake. It appears that the 6AzGal uptake assay is acting in a similar fashion to 2NBDG in this instance... therefore how is this to be a trusted uptake assay??

~ Perhaps consistent use of transporter inhibitors is required as a vital control in each assay?

Response #2-7: We guess that the reviewer misunderstood the relevance of our work and the previous report (*Immunometabolism* 2, e200029 (2020)). Our data of thymocyte (Fig. 4d) were obtained by *in vivo* analysis using fasted mice, whereas the previous study does not conduct any *in vivo* analysis but

only *ex vivo* analysis. The previous study is simply focused on comparison of the responses on 2DG and 2NBDG uptake between the isolated cells and activated T cells. Furthermore, our results showing increased 6AzGal uptake in activated T cells (Fig. 4c) were well consistent with these previous studies (*Immunometabolism* 2, e200029 (2020); *Cell Metab.* 20, 61 (2014)). As the reviewer commented, GLUT inhibitors may be useful for *in vivo* analysis. However, it is not clear that the intended effect in the target tissue can be adequately obtained, because *in vivo* dynamics of sugars and inhibitors are not always consistent.

In working up this assay, the authors should be commended on their use of transporter controls, and comparisons with valid glucose uptake assays, as well as comparisons with existing flawed assays which they are trying to better. However, it is clear from the data presented that much more is needed to be known about the transport parameters of 6AzGal. Furthermore, the output or “uptake measurement” CANNOT be governed by the variability in the labelling of the detection reagent.

Response #2-8: A number of sugar-analog probes for monitoring GLUT-mediated glucose uptake have been developed to date (*Nat. Commun.* 13, 5974 (2022); *Anal. Chem.* 94, 8293 (2022); *Chem. Commun.* 56, 1070 (2020); *Nat. Methods* 16, 526 (2019); *Bioconjug. Chem.* 29, 3394 (2018); *Biochem. Biophys. Res. Commun.* 480, 341 (2016); *Biochem. Biophys. Res. Commun.* 474, 240 (2016); *Angew. Chem.* 54, 9821 (2015); *Org. Biomol. Chem.* 9, 4760 (2011); *Angew. Chem.* 48, 8027 (2009) etc.). In a similar way to these published studies, our method was extensively validated by using competitive inhibitors, GLUT-selective inhibitors, activated cells, animal models, and other probes. Given that our work comprehensively covers these standard validation experiments, we do not judge that there is a serious lack of scientific evidence for qualifying our method. We believe that the more in-depth mechanistic study is out of the scope of this manuscript.

The authors need to nail down these issues in order for this assay to be any better than the flawed assays already in existence. Specifically – demonstrably linear uptake parameters which conform to physiological transporter parameters AND the output should not be reliant on background correction which is highly variable in the absence of the uptake substrate.

Response #2-9: As described in **Response #2-1**, we conducted the concentration-response experiments (Fig. S2i), and obtained a histogram of 6AzGal-dilution samples (left panel in Fig. S2i), presenting the raw MFI data as a graph with a high linearity ($R^2 = 0.95$) (center panel in Fig. S2i). By comparing MFIs of different 6AzGal concentration samples to that of no-azide control (0 mM 6AzGal), MFI fold changes were also determined and again presented as a graph with a good linearity ($R^2 = 0.94$) (right panel in Fig. S2i). Regardless of the background correction, our data provided these linear outputs in a range of 0-20 mM 6AzGal concentrations. These indicate our assay method to be robust and highly accurate. Because raw fluorescence intensities in flow cytometry could vary depending on

the machine and sample conditions, we used MFI fold changes to minimize the day-to-day variation in this study. Alternatively, evaluation based on MFI differences or cell populations might be used depending on research purposes and applications of various researchers.

Reviewers' comments:

Reviewer #1 (Remarks to the Author):

The authors have meticulously addressed nearly all the raised queries; however, I have a few trivial requests outlined below:

1. Kindly ensure the appropriate utilization of standard Greek letters for unit representation. In certain main Figures (e.g. Fig 3d, 3e, etc.) the micro- symbols are denoted as "u," and this ought to be converted to the Greek letter "μ" (mu).
2. It is requested that the authors provide further details, either in the main text or supporting information, elucidating the methodology employed to obtain the linear fit and the corresponding R2 value for each graph (Fig.S2i, S2j). For instance, in the uptake kinetics plot of 6AzGal in K562 cells, the trend line is reported to have an R2 value of 0.99, despite depicting a plateau over time. It is imperative to elucidate the range of data points utilized to derive the R2 value, and a clear explanation of the obtained value and fitting methodology is essential.

The revised manuscript is on the verge of being publication-ready following these minor refinements.

Reviewer #2 (Remarks to the Author):

I would very much like to thank the authors for their clear, polite and comprehensive approach in addressing the points that both myself and the other reviewer(s) raised on the initial submission of their paper.

They really have addressed all the serious concerns that I had.

For my part of the review, I am pleased that the authors clarified many key points for me. I do however think that if I managed to misinterpret these key points then it may well be expected for many in the community (ie the flow-centric immunology field!) would also have these same problems.

I would strongly ask that the authors think about including a "dummies guide" model of EXACTLY how to set-up the "perfect" uptake and staining protocol, including washes and buffers, AND the optimal controls needed for accurate interpretation of the data.

I realise this is within the methods section, but a simple "go-to" flow diagram including the importance of the controls would be very helpful for the community at large.

Finally –though in depth investigation of this is well outside the purview of this paper –I feel that it IS of interest to discuss the high probe uptake seen in the thymocytes.

I have attached screen shot of Slc2a1 RNA expression in thymocytes (from the immgen database), a screenshot of radiolabelled 2DG uptake in total thymocytes (from Sinclair et al Immunometabolism 2, (2020) -ref11 in the manuscript) and a screenshot of radiolabelled 2DG uptake in DN3, DN4 and DP thymocyte subsets (from fig3 Swamy et al Nat Immunol 17, 712-720 (2016) doi: 10.1038/ni.3439)

I have also attached screenshots from the current manuscript: supplemental in vivo data showing 6AzGal uptake from splenic T cells

Fig4 in vivo data showing 6AzGal uptake from whole thymus
AND, a crude overlay of the two- with the splenic T cells in reverse colour.

Here is why I am intrigued by the high probe uptake in whole population thymocytes -- as a "total" population:

Thymocytes are approx. 80-90% DP cells. Very few DNs (normally).

Immgen RNA data shows that slc2a1 expression is lowest in the most abundant thymocyte population (DPs) and is increased in the much less abundant precursor DN1, DN3, ISP and DN4 cells.

Furthermore whole thymocyte populations do NOT take up high levels of 3H 2-DG.

When broken down into sub-populations, DP thymocytes do NOT take up high levels of 3H 2-DG, with highest levels of uptake seen in DN4 cells.

These data correlate strongly with the Immgen RNAseq data.

- Yes, I realise that these data are all ex vivo, and not in vivo as the paper shows...

BUT the in vivo labelling data in the manuscript appears to show HIGHER labelling of thymocytes than naïve T cells from the spleen. (Even higher than the B cells in the spleen, which the authors show have higher uptake than T cells....)

This is my conundrum ...

- This uptake assay undoubtedly implicates high uptake into thymocytes,
- however to my knowledge, DP cells do not express high levels of SLC2A1 or 3 or other known glucose - galactose- transporters.
- NOR do ex vivo radiolabelled uptake assays support the notion that total thymocytes have high levels of glucose (2DG) transport.

So how can one trust that this transporter reports (only) on glucose transport capacity?

And this is what the general user will always interpret ...

I know that this may seem to be a minor point, but I do believe this assay has the potential to be very well-used - that is why I am "harping" on about it!

The diligence that the authors have shown with the controls throughout the paper is truly commendable. However, a paragraph/discussion sentence addressing HEAD - ON this current enigma (who knows perhaps someone WILL find a weird and wonderful galactose not glucose transporter in thymocytes!) would be beneficial.

(I do apologise for the long-winded way of asking for this!)

(I am hoping that the editor will forward the images, as I am aware that they have not dropped into this text box)

Point-by-Point Responses to Reviewers

The followings are our responses (highlighted in BLUE) to all of referees' comments.

Reviewers' comments:

Reviewer #1 (Remarks to the Author):

The authors have meticulously addressed nearly all the raised queries; however, I have a few trivial requests outlined below:

We thank for the Reviewer's requests and addressed them as follows.

1. Kindly ensure the appropriate utilization of standard Greek letters for unit representation. In certain main Figures (e.g. Fig 3d, 3e, etc.) the micro- symbols are denoted as "u," and this ought to be converted to the Greek letter " μ " (mu).

Response #1-1: We appropriately utilized the Greek letter " μ " in Fig. 3e (uM changed to μ M).

2. It is requested that the authors provide further details, either in the main text or supporting information, elucidating the methodology employed to obtain the linear fit and the corresponding R2 value for each graph (Fig.S2i, S2j). For instance, in the uptake kinetics plot of 6AzGal in K562 cells, the trend line is reported to have an R2 value of 0.99, despite depicting a plateau over time. It is imperative to elucidate the range of data points utilized to derive the R2 value, and a clear explanation of the obtained value and fitting methodology is essential.

Response #1-2: The linear fitting and the R2 values were obtained using Microsoft Excel. This information was described in Statistics and Reproducibility in Method section. Related data and calculation were present in Source Data file (Excel).

The revised manuscript is on the verge of being publication-ready following these minor refinements.

Reviewer #2 (Remarks to the Author):

I would very much like to thank the authors for their clear, polite and comprehensive approach in addressing the points that both myself and the other reviewer(s) raised on the initial submission of their paper.

They really have addressed all the serious concerns that I had.

For my part of the review, I am pleased that the authors clarified many key points for me. I do however think that if I managed to misinterpret these key points then it may well be expected for many in the community (ie the flow-centric immunology field!) would also have these same problems.

We thank the reviewer for the positive assessment of this work. We addressed the comments as follows.

I would strongly ask that the authors think about including a “dummies guide” model of EXACTLY how to set-up the “perfect” uptake and staining protocol, including washes and buffers, AND the optimal controls needed for accurate interpretation of the data. I realise this is within the methods section, but a simple “go-to” flow diagram including the importance of the controls would be very helpful for the community at large.

Response #2-1: As we previously published our labeling techniques in STAR Protocols (doi: 10.1016/j.xpro.2023.102525) and Current Protocols (doi: 10.1002/cpz1.105), we will plan to submit a detailed protocol paper on this 6AzGal uptake assay in an appropriate journal.

Finally –though in depth investigation of this is well outside the purview of this paper –I feel that it IS of interest to discuss the high probe uptake seen in the thymocytes.

I have attached screen shot of Slc2a1 RNA expression in thymocytes (from the immgen database), a screenshot of radiolabelled 2DG uptake in total thymocytes (from Sinclair et al Immunometabolism 2, (2020) -ref11 in the manuscript) and a screenshot of radiolabelled 2DG uptake in DN3, DN4 and DP thymocyte subsets (from fig3 Swamy et al Nat Immunol 17, 712-720 (2016) doi: 10.1038/ni.3439)

I have also attached screenshots from the current manuscript:
 supplemental in vivo data showing 6AzGal uptake from splenic T cells
 Fig4 in vivo data showing 6AzGal uptake from whole thymus
 AND, a crude overlay of the two- with the splenic T cells in reverse colour.

Here is why I am intrigued by the high probe uptake in whole population thymocytes — as a “total” population: Thymocytes are approx. 80-90% DP cells. Very few DNs (normally). Immgen RNA data shows that *slc2a1* expression is lowest in the most abundant thymocyte population (DPs) and is increased in the much less abundant precursor DN1, DN3, ISP and DN4 cells. Furthermore whole thymocyte populations do NOT take up high levels of 3H 2-DG. When broken down into sub-populations, DP thymocytes do NOT take up high levels of 3H 2-DG, with highest levels of uptake seen in DN4 cells. These data correlate strongly with the Immgen RNAseq data.

- Yes, I realise that these data are all ex vivo, and not in vivo as the paper shows...

BUT the in vivo labelling data in the manuscript appears to show HIGHER labelling of thymocytes than naïve T cells from the spleen. (Even higher than the B cells in the spleen, which the authors show have higher uptake than T cells....)

This is my conundrum ...

- This uptake assay undoubtedly implicates high uptake into thymocytes,
- however to my knowledge, DP cells do not express high levels of SLC2A1 or 3 or other known glucose – galactose- transporters.
- NOR do ex vivo radiolabelled uptake assays support the notion that total thymocytes have high levels of glucose (2DG) transport.

So how can one trust that this transporter reports (only) on glucose transport capacity?

And this is what the general user will always interpret ...

I know that this may seem to be a minor point, but I do believe this assay has the potential to be very

well-used - that is why I am “harping” on about it!

The diligence that the authors have shown with the controls throughout the paper is truly commendable. However, a paragraph/discussion sentence addressing HEAD – ON this current enigma (who knows perhaps someone WILL find a weird and wonderful galactose not glucose transporter in thymocytes!) would be beneficial.

(I do apologise for the long-winded way of asking for this!)

Response #2-2: According to the Reviewer’s request, we describe potential explanations for a difference between our *in vivo* 6AzGal data and the *ex vivo* 3H-2DG data. First of all, we are afraid that the reviewer may overinterpret our *in vivo* data. Because the main scope of this paper is demonstration of the *in vivo* use of 6AzGal, we set the experimental conditions of Fig. 4d to qualitatively detect 6AzGal uptake in each tissue, but not to quantitatively compare 6AzGal uptake activities among these tissues (e.g. spleen vs thymus). Due to passive diffusion of BDP-DBCO across cell membranes, fluorescence labeling yield can be varied in independently-prepared tissue samples (*Cell Metab.* 35, 1072 (2023); *STAR Protoc.* 4, 102525 (2023)), which needs use of fluorescence barcodes to simultaneously label multiple samples in a pooled format for accurate comparison of azide content. Given that the data set presented in Fig. 4d was obtained through different samples without barcoding, additional *in vivo* validation for the comparative 6AzGal analysis (spleen vs thymus) should be required, for example by using 18F-FDG with PET imaging or gamma emission recording (X Xiang, *Sci. Transl. Med.* 13, eabe5640 (2021)). However, as the reviewer agrees that this in-depth investigation is well out of the scope of this manuscript, we would like to describe a possible mechanistic difference between 6AzGal and 3H-2DG as follows. Unlike 3H-2DG, 6AzGal is neither phosphorylated at the 6-position by hexokinase nor at the at the 1-position by galctokinase (*ACS Chem. Biol.* 15, 318 (2020); *Biochem. Soc. Trans.* 44, 116 (2016); *J. Biol. Chem.* 280, 9662 (2005)). Given that the hexokinase-dependent phosphorylation is critical for intracellular accumulation of 3H-2DG but not for that of 6AzGal, an uptake signal of 6AzGal does not perfectly match that of 3H-2DG. Moreover, overnight-fasted mice were used in our *in vivo* 6AzGal analyses like the previous 18F-FDG imaging study (*J. Nucl. Med.* 52, 800 (2011)). This abstaining from food should impact on metabolic states and flows in the tissues, while the *ex vivo* 3H-2DG studies used normal fed mice. Therefore, we consider that the different biological situations (in vivo vs ex vivo and fasting vs fed) might affect thymic/splenic glucose uptake. These comments are added in the discussion part.

REVIEWERS' COMMENTS:

Reviewer #2 (Remarks to the Author):

The authors have addressed all my queries, thank you. I look forward to seeing this paper in print!